# Syphilis and Co-Infections with HIV-1, HBV, and HCV among People Who Use Crack-Cocaine in Northern Brazil

**DOI:** 10.3390/pathogens11091055

**Published:** 2022-09-16

**Authors:** Karen Lorena N. Baia, Ana Caroline C. Cordeiro, Paula Cristina R. Frade, Alanna Gabrielly N. Gouveia, Rafael Lima Resque, Luiz Marcelo L. Pinheiro, Ricardo Roberto S. Fonseca, Luiz Fernando A. Machado, Luisa C. Martins, Emil Kupek, Benedikt Fischer, Aldemir B. Oliveira-Filho

**Affiliations:** 1Programa de Pós-Graduação em Doenças Tropicais, Universidade Federal do Pará, Belém 66075-110, Brazil; 2Residência Multiprofissional em Saúde da Mulher e da Criança, Hospital Santo Antônio Maria Zaccaria, Bragança 68600-000, Brazil; 3Grupo de Estudo e Pesquisa em Populações Vulneráveis, Instituto de Estudos Costeiros, Universidade Federal do Pará, Bragança 68600-000, Brazil; 4Departamento de Ciências Biológicas e da Saúde, Universidade Federal do Amapá, Macapá 68903-419, Brazil; 5Faculdade de Ciências Biológicas, Campus do Marajó, Universidade Federal do Pará, Soure 68870-000, Brazil; 6Laboratório de Virologia, Instituto de Ciências Biológicas, Universidade Federal do Pará, Belém 66077-830, Brazil; 7Laboratório de Patologia Clínica de Doenças Tropicais, Núcleo de Medicina Tropical, Universidade Federal do Pará, Belém 66055-240, Brazil; 8Departamento de Saúde Pública, Universidade Federal de Santa Catarina, Florianópolis 88040-900, Brazil; 9Centre for Applied Research in Mental Health and Addiction, Faculty of Health Sciences, Simon Fraser University, Vancouver, BC V6B 5K3, Canada; 10Faculty of Medical and Health Sciences, University of Auckland, Auckland 1023, New Zealand; 11Departamento de Psiquiatria, Universidade Federal de São Paulo, São Paulo 04038-000, Brazil

**Keywords:** syphilis, co-infections, crack-cocaine use, epidemiology, public health, interventions, Brazil

## Abstract

The rates of syphilis and viral co-infections among people who use crack-cocaine (PWUCC) were assessed in this study. This cross-sectional study relied on biological and self-reported socio-behavioral data from a convenience sample of 990 PWUCC from twenty-six municipalities in the states of Amapá and Pará, northern Brazil. Blood samples were collected to assess the presence of *Treponema pallidum* using the Rapid Qualitative Test (RQT) and the Venereal Disease Research Laboratory (VDRL). Reactive samples by RQT were used to assess the presence of HBV, HCV, and HIV-1 using Enzyme Immunoassay (EIA) and Polymerase Chain Reaction (PCR). Logistic regression models were used to determine the association of variables assessed with syphilis. In total, 287 (29.0%) of the PWUCC sample had reactive results for syphilis. HBV (15.7%), HCV (5.9%), and HIV-1 (9.8%) were detected among PWUCC with syphilis. Young age, low monthly income and education level, long duration of crack-cocaine use, condomless sex, multiple sex partners, and exchange of sex for money/drugs were associated with syphilis. The present study provides unique insights on the epidemiological status of syphilis among PWUCC in northern Brazil, with multiple implications for improving urgent interventions for diagnosis, prevention, and treatment.

## 1. Introduction

South America is home to the extensive production and consumption of cocaine and cocaine-based psychoactive drug products [1]. Crack-cocaine is made from cocaine base obtained from cocaine hydrochloride through conversion processes to make it suitable for smoking [1,2]. In Brazil, crack-cocaine use has received greater attention from citizens and governments due to its expanding markets, with frequent use of this substance among the homeless and impoverished [2,3,4]. Its consumption has been associated with different adverse physical and psychiatric health conditions, violence, and high rates of mortality [3,5,6,7,8]. Additionally, crack-cocaine use has shown a high association with sexually transmitted infections (STIs), mainly due to behaviors such as condomless sex and multiple sex partners [3,9,10,11,12,13].

Among STIs, syphilis is an important public health problem worldwide. It is caused by *Treponem pallidum*, a motile Gram-negative spirochaete [14]. Syphilis is a vertical, blood-related, systemic infectious disease with chronic evolution, and *T. pallidum* is transmitted mainly by sexual contact. This spirochete is not evenly distributed in the population. The transmission rates are related to a range of social-behavioral biological and cultural factors that influence the occurrence of infections in the population [15,16]. Men who have sex with men (MSM), transgender persons (TGP), female sex workers (FSW), and people who use illicit drugs (PWUD) are considered high-risk groups for STIs such as syphilis [14,15,16,17,18,19]. In Brazil, high prevalence levels of syphilis (4.0–29.8%) have been reported among people who use crack-cocaine (PWUCC). Low education levels, condomless sex, exchange of sex for money and/or drugs, and history of genital ulcers have been factors associated with syphilis in this high-risk group [7,20,21,22].

In recent years, epidemiological studies conducted with PWUD, including PWUCC, in northern Brazil have detected high rates of viral infections and risk factors associated with drug use (e.g., daily drug use), sexual behaviors [e.g., exchange of sex for money and/or drugs], and social vulnerability (e.g., poverty and unstable housing, including homelessness) [11,12,13,23,24,25]. This Brazilian region, which includes most of the tropical Amazon rainforest, can be characterized as a rural and socioeconomically underdeveloped area, with high levels of poverty, limited transport infrastructure, and inadequate health services [11]. Epidemiological reports have highlighted the urgent need for public health institutions in northern Brazil to better serve and care for vulnerable and marginalized populations, such as PWUCC, especially in the context of the diagnosis and treatment of STIs and treatment for risky drug use and disorders [12,13,23,26]. Thus, this study aimed to assess the rates of syphilis and co-infections with hepatitis B virus (HBV), hepatitis C virus (HCV), and human immunodeficiency virus type 1 (HIV-1) among PWUCC in northern Brazil, as well as identifying the factors associated with infection by this spirochete.

## 2. Results

### 2.1. Study Sample

This study consisted of both biological samples and questionnaire data on individual characteristics from 990 PWUCC, 197 of which were accessed in six municipalities in the state of Amapá and 793 in 20 municipalities in the state of Pará (Figure 1). In the snowball chain length, the average number of PWUCC in each municipality was 38 (standard deviation = ±37). The highest and lowest number of PWUCC was obtained in Bragança (n = 165) and Ponta de Pedras (n = 9), respectively (Appendix A).

### 2.2. Characteristics of PWUCC

The sample’s average age was 28.2 years (±7.3 years). Most PWUCC were male, single, heterosexual, had a low educational level, low monthly income, reported condomless sex, had up to 10 sexual partners, and did not access public health services in the last 12 months (Table 1). Some participants reported the exchange of sex for money or illicit drugs (n = 303; 30.6%).

The average length of participants’ crack-cocaine use was 39.2 months (±22.5 months), and the daily amount of crack-cocaine averaged 14.2 “rocks” (±8.1 stones). The use of pipes, cans, and other plastic or metal reservoirs (including equipment made manually by the participant) was observed for crack-cocaine use, and the sharing of equipment for drug use has been reported (33.0%) (Appendix A). Some participants also reported the use of crack-cocaine combined with marijuana (14.5%). The co-use of alcohol, tobacco, and inhalant drugs was reported by half of the PWUCC (51.1%).

### 2.3. Syphilis and Viral Co-Infections

Overall, 287 (29.0%) had reactive results for syphilis. Forty-nine (4.9%) PWUCC had reactive results for treponemal antibodies using Rapid Qualitative Test (RQT) and non-reactive for non-treponemal antibodies using Venereal Disease Research Laboratory (VDRL). Two hundred and thirty-eight (24.0%) PWUCC had reactive results for both treponemal and non-treponemal antibodies (Table 2). Most of them showed high titration values using VDRL and reported the presence of lesions on the skin (n = 78) or the oral/genital mucosa (n = 143) (Appendix A). In addition, 90 (31.4%) PWUCC with syphilis had viral co-infections (Table 2).

In this sample, forty-five (15.7%) PWUCC were exposed to HBV, of whom four tested positive for HBsAg, and forty-one had anti-HBc. Six PWUCC exposed to HBV had virus nucleic acid. Regarding HCV, seventeen (5.9%) PWUCC tested positive for anti-HCV antibodies, and eight of them had HCV RNA. In addition, twenty-eight PWUCC reactive results for syphilis tested positive for anti-HIV-1/2 antibodies, and nineteen of them had HIV-1 RNA. Finally, eleven (2.7%) PWUCC showed positive results for syphilis, HBV, and HIV-1 (Table 2). All PWUCCs who tested positive only for syphilis or syphilis and co-infection with HBV, HCV, and/or HIV-1 reported not being aware of their carrier status of these pathogens until participating in this study, consequently, none of them had received treatment for any of the pathogens.

### 2.4. Factors Associated with Syphilis

Logistic regression models identified seven factors associated with syphilis among PWUCC. Bivariate and multivariate analyses indicated the following factors: young (18–29 years), low education level (up to elementary school), low monthly income (up to monthly Brazilian minimum wage), long history of crack-cocaine use (crack-cocaine use ≥40 months), condomless sex, more than 10 sexual partners, and exchange of sex for money or illicit drug (Table 3). The Hosmer–Lemeshow test indicated that the final model (_HL_χ^2^ = 10.4; *p* = 0.25) had a good fit. The factors most strongly associated with syphilis were “condomless sex” (aOR = 18.3) and “more than 10 sexual partners in the last 12 months” (aOR = 4.7) (Table 3). The factors whose association with syphilis did not reach the statistical significance of *p* < 0.05 are shown in the Appendix A.

## 3. Discussion

Important findings from the epidemiological scenario of syphilis among PWUCC in northern Brazil are presented from this epidemiological study, making this the first with a specific target population in the Amazon region. In the sample, positivity rate for syphilis was 29%, as well as it was discovered that up to one in three of the PWUCC diagnosed with syphilis was co-infected with HIV-1, HBV, or HCV. This rate of syphilis is similar to levels found in the PWUCC sample (n = 1,062) in the Brazilian state of Pernambuco (29.8%). On the other hand, the prevalence for HBV, HCV, and HIV recorded here were somewhat higher than the findings among PWUCC in Pernambuco (HBV = 4.3%; HCV = 1.6%; HIV = 6.9%) [7]. The prevalence of syphilis among PWUCC is comparably high when compared to findings in several vulnerable and marginalized groups in Brazil, such as people living with HIV/AIDS (6.4%), homeless men (7–10.2%), MSM (9.9%), and prisoners (10.5–22.1%), and other South American countries—MSM and TGP [14,27,28,29,30,31]. However, studies conducted with FSW in northern Brazil (36.1–41.1%) and Argentina (30.3–51.5%) reported a much higher prevalence of syphilis than the levels detected in the present study [15,19,32,33].

PWUCC and FSW may have very similar socio-behavioral risk factor contexts and environments, as well as the main characteristics related to the use of drugs and behaviors relevant to the acquisition of *T. pallidum* observed in this study. This becomes more evident when considering viral co-infections. High rates of exposure to HBV, HCV, and HIV were detected in the present PWUCC sample. In the Marajó Archipelago (northern Brazil), high rates of exposure to HBV (23.0%) and HCV (8.1%) were previously detected with FSW with syphilis. In these women, several risk factors were associated with syphilis: low education level, low income, drug use, condomless sex, a lengthy sex work history, and a lack of medical care access [19].

In the present study, several independent risk factors for syphilis were determined in the PWUCC sample using multivariate analysis. They can be organized into three correlated groups. First, engagement in condomless sex was recorded as the strongest risk factor. Other sexual behavior-related factors were also identified with lower OR values (e.g., high number of sexual partners and exchange of sex for money/illicit drugs). This provides clear evidence of the fact that PWUCC acquired *T. pallidum* through sexual contact pathways. Condomless sex and other sex-related factors have been found to contribute to the acquisition and spread of *T. pallidum* in other population groups such as FSW, MSM, and prisoners [29,33,34]. Second, a long-term history of crack-cocaine use was one of the factors associated with syphilis. Long-term crack-cocaine use can greatly increase the risk of acquiring pathogens because PWUCC increasingly engage in risk situations and consequently decreased health status and care. The presence of HBV, HEV, human papillomavirus (HPV), HIV-1, human T-lymphotropic virus 1 (HTLV-1), and 2 (HTLV-2) has been recorded among PWUCC who use this psychoactive substance for extended periods and engage in risk situations, such as condomless sex and the sharing of crack-cocaine equipment [9,10,11,13,23,24]. Third, the predictive role of lower educational levels, low monthly income, and younger age as factors associated with syphilis points to the role of adverse socioeconomic determinants and specifically to social marginalization. In addition to the initiation of crack-cocaine use, socioeconomic factors also influence the severity of behaviors and outcomes associated with drug use and users’ access to care [35,36,37]. In this context, the occurrence of situations that facilitate the acquisition and transmission of pathogens is well-documented among PWUCC compared to other subpopulations of PWUD [6,11,20,38].

The results also point to interventions in the epidemiological scenario. No PWUCC with syphilis were aware of their infection status and, consequently, were not undergoing any standard treatment [39]. This situation urgently needs to be addressed and improved. It is essential to identify and treat people infected with pathogens, as well as thereby preventing future infections in others. Improvement of resources, services, and technical skills must be carried out in public health establishments in this Brazilian region. In addition, the primary risk factor for syphilis is related to sexual encounter; therefore, there is a need to improve prevention measures. Targeted education programs, provision of resources or prevention materials (e.g., condoms), and a clear understanding of the association of condomless sex and the economy or acquisition of drugs, and other more specific measures related to the local population are important prevention measures to reduce the spread of pathogens among PWUD through sexual contact [11,13]. Finally, interventions should target the issue of crack-cocaine use itself, given the duration and intensity of use of this psychoactive substance in the target population. Most participants require evidence-based treatment for (multi-)drug dependence/disorder [40,41]. The availability of resources to host and care for PWUD are highly limited in northern Brazil [11,12,13]. The development of intervention measures related to the use of psychoactive substances is very important for the treatment of drug dependence/disorder and to contain or reduce the spread of pathogens in the present target population.

This study has some limitations which require consideration. The PWUCC sample assessed consisted of a convenience sample and, therefore, is not generalizable to other populations of PWUCC or illicit drug users. This study also did not include clinical assessments of PWUCC, which may be relevant to the status of active syphilis. The participants with reactive results for RQT and non-reactive for VDRL were not evaluated by another treponemal test, such as fluorescent treponemal antibody absorption test (FTA-Abs), as suggested by the Brazilian Ministry of Health [42]. Self-report approaches were used in this study to record epidemiological factors, which are not objectively verifiable and may include multiple biases. Finally, the cross-sectional design of the study limits its capacity to establish causality of outcomes.

## 4. Materials and Methods

### 4.1. Study Design and Sampling

This cross-sectional study relied on biological and self-reported socio-behavioral data (including knowledge about the presence of HBV, HCV, HIV-1, and *T. pallidum* and treatment of infections caused by these pathogens) from a convenience sample of PWUCC from the municipalities of the state of Amapá and Pará, northern Brazil (Figure 1). Sample recruitment occurred by way of the snowball technique in epidemiological studies conducted in the two Brazilian states [11,12,13]. Study eligibility criteria were: (1) use of crack-cocaine in the last three months; (2) 18 years of age or older; (3) not under the effect of psychotropic drugs during the meeting with some members of the study team; (4) did not present a risk of death to researchers in the municipality; (5) willingness to comply with the study protocol, i.e., to provide a biological sample, to complete the epidemiological assessment, and to provide informed consent for study participation.

### 4.2. Sample Collection and Laboratory Tests

At all study sites, blood samples collected from the PWUCC were examined using RQT for syphilis (Bioline™ SYPHILIS 3.0, Standard Diagnostics Inc., Yongin-si, Korea), according to manufacturer’s guidelines. RQT for syphilis are considered treponemal tests as they involve the detection of specific antitreponemal antibodies in a sample [42]. In this study, PWUCC with non-reactive results by RQT were classified as without syphilis and no procedure or test was performed. All PWUCC with reactive results by RQT were classified as with syphilis (i.e., they have been (past) or are (present) infected with *T. pallidum*) and had biological samples collected to perform additional tests (Figure 2). Blood samples (5 mL) from PWUCC with syphilis were also collected by venipuncture with disposable syringes and tubes containing anticoagulants. Each sample was centrifuged at room temperature; the resulting plasma was stored at a temperature between 4 and 8 °C and transported to the laboratory. Reactive samples by RQT were tested by VDRL (VDRL/Sífilis, WAMA Diagnóstica, Brazil). VDRL is considered a non-treponemal test, as it detects antibodies that are not specific for *T. pallidum*, but that are found in patients with syphilis [42]. All samples were tested pure (1:1) and in titrations (≥1:2) to eliminate the possibility of the prozone phenomenon in the execution of this laboratory test [42]. Reactive samples by RQT were used to assess exposure to HBV (HbsAg–AxSYM HbsAg, Abbott, Chicago, IL, USA; Anti-HBc-Total Murex Anti-HBc, DiaSorin, Italy), HCV (Murex anti-HCV 4.0, DiaSorin, Saluggia, Italy), and HIV-1 (Murex HIV-1.2.O kit; DiaSorin, Saluggia, Italy) using enzyme immunoassay (EIA). The confirmation of these viral infections was made using protocols and commercial kits for real-time Polymerase Chain Reaction (PCR) [11,12,43]. All PWUCC biological samples and personal data were collected from March 2013 to November 2018.

### 4.3. Data Collection and Statistical Analysis

All PWUCC completed an interviewer-administered questionnaire consisting of question items on socio-demographics, drug use, and select clinical/health and behavior outcome variables used in previous studies with similar target populations [11,12,43].

These data were fed into an Excel database and converted to SPSS format for all procedures and statistical analyses. The 95% confidence intervals (CI) were determined to estimate the rate of syphilis and co-infections with different viruses. A descriptive analysis was conducted to investigate the bivariate relationships between syphilis (outcome) and epidemiological (i.e., socio-demographic, drug use, and other health-related) covariates drawn from the epidemiological questionnaire data. All the potential factors with probabilities of *p* ≤ 0.2 were examined and included in the final model of analysis on the outcomes of syphilis, using backward stepwise multiple logistic regression. Multiple logistic regressions were then run to determine the association of each factor with the outcome. Various possible types of interactions were evaluated to determine how they might influence the final model. The overall fit of the final model was assessed using the Hosmer–Lemeshow test. The latter used a 0.05 significance level for the type I error. The data were analyzed using SPSS Statistics 23.0 (IBM, Armonk, NY, USA).

## 5. Conclusions

The present study provides unique insights on the status of syphilis and risk factors among PWUCC in a multi-site sample in northern Brazil, with multiple implications for improving on urgent interventions for diagnosis, prevention, and treatment. The lack of services and measures to provide targeted prevention and care for PWUCC in northern Brazil negatively impacts the entire community, through the transmission of viral pathogens without effective measures to control and provide treatment to infected people, especially those at elevated risk of acquisition and transmission.

## Figures and Tables

**Figure 1 pathogens-11-01055-f001:**
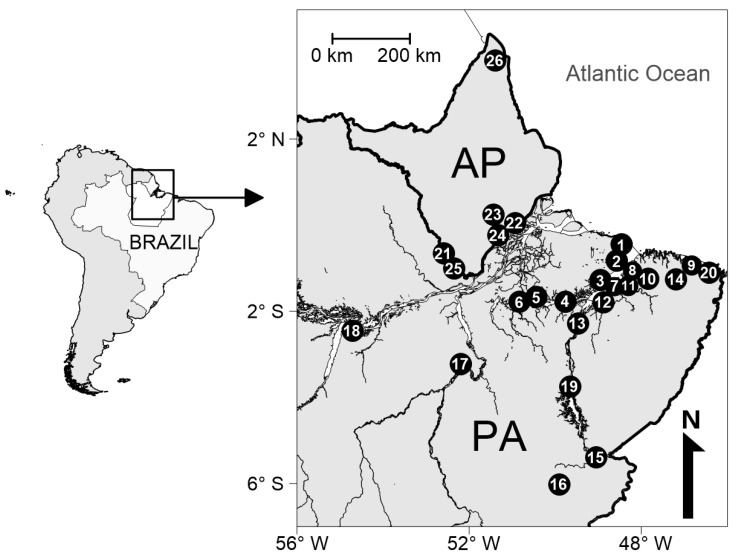
Location of the municipalities from where people who use crack-cocaine were accessed in in the states of Amapá (AP) and Pará (PA), northern Brazil. The numbers from 1 to 26 indicate the municipalities: Soure (1), Salvaterra (2), Ponta de Pedras (3), Curralinho (4), Breves (5), Melgaço (6), Belém (7), Benevides (8), Bragança (9), Castanhal (10), Marituba (11), Abaetetuba (12), Cametá (13), Capanema (14), Marabá (15), Parauapebas (16), Altamira (17), Santarém (18), Tucuruí (19), Augusto Corrêa (20), Laranjal do Jari (21), Macapá (22), Mazagão (23), Santana (24), Vitória do Jari (25), and Porto Grande (26). More details in the Appendix A.

**Figure 2 pathogens-11-01055-f002:**
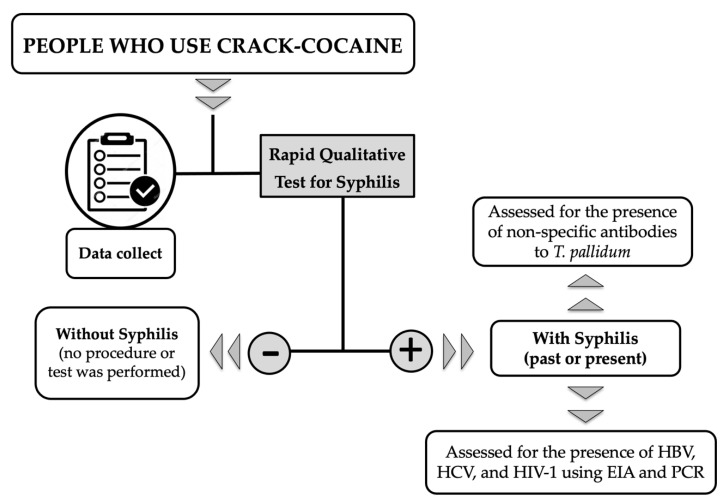
Flowchart of actions and laboratory tests used in this study. VDRL: Venereal Disease Research Laboratory; EIA: Enzyme immunoassay; PCR: Polymerase Chain Reaction; HBV: Hepatitis B virus; HCV: Hepatitis C virus; HIV-1: Human immunodeficiency virus type 1.

**Table 1 pathogens-11-01055-t001:** Socio-demographic, economic, and behavioral characteristics of people who use crack-cocaine related to syphilis in northern Brazil.

Characteristics	All	Syphilis + (%)	Syphilis − (%)	*p*-Value *
Total	990	287 (29.0)	703 (71.0)	-
Sex				
Male	720	199 (27.6)	521 (72.4)	0.13
Female	270	88 (32.6)	182 (67.4)	
Age (years)				
18–29	632	214 (33.9)	418 (66.1)	<0.01
30–40	323	67 (20.7)	256 (79.3)	
>40	35	6 (17.1)	29 (82.9)	
Color/race (self-identified)				
White	119	37 (31.1)	82 (68.9)	0.09
Pardo (mixed race)	594	170 (28.6)	424 (71.4)	
Black	277	80 (28.9)	269 (71.1)	
Marital status ^†^				
Single	785	231 (29.4)	554 (70.6)	0.83
Separated or widowed	148	40 (27.0)	108 (73.0)	
Married or co-habiting	57	16 (28.1)	41 (71.9)	
Educational level				
No formal education	88	26 (29.5)	62 (70.5)	0.01
Up to elementary school	703	220 (31.3)	483 (68.7)	
Up to high school/postgraduate	199	41 (20.6)	158 (79.4)	
Monthly income (Brazilian minimum wage) ^†^				
≤ one minimum wage ^‡^	750	242 (32.3)	508 (67.7)	<0.01
2–3 times the minimum wage	190	43 (22.6)	147 (77.4)	
>3 times the minimum wage	50	2 (4.0)	48 (96.0)	
Housing status ^†^				
Live in your own home or with parents	556	154 (27.7)	402 (72.3)	
Lives in house or rented room	277	81 (29.2)	196 (70.8)	0.42
Unstable housing	157	52 (33.1)	105 (66.9)	
Crack-cocaine use time (months)				
Up to 19	119	17 (14.3)	102 (85.7)	<0.01
20–39	290	82 (28.3)	208 (71.7)	
>40	581	188 (32.4)	393 (67.6)	
Shared use of crack-cocaine equipment ^†^				
Yes	327	106 (32.4)	221 (67.6)	0.10
No	661	181 (27.4)	482 (73.6)	
Sexual orientation				
Heterosexual	901	264 (29.3)	637 (70.7)	0.49
Same sex (including bisexual)	89	23 (25.8)	66 (74.2)	
Condom use during sex ^†^				
Rarely + Never	697	218 (31.3)	479 (68.7)	<0.01
Sometimes	178	63 (35.4)	115 (64.6)	
Always	115	6 (5.2)	109 (94.8)	
Number of sexual partners ^†^				
More than 10	451	149 (33.0)	302 (77.0)	0.01
Up to 10	539	138 (25.6)	401 (74.4)	
Oral sex ^†^	602	181 (30.1)	421 (69.9)	0.35
Anal sex^†^	311	98 (31.5)	213 (78.5)	0.24
Exchange of sex for money or illicit drug ^†^	303	115 (38.0)	188 (62.0)	<0.01
Did not access public health service ^†^	902	269 (29.8)	633 (70.2)	0.06

* Calculated by Chi-square test. ^†^ Last 12 months; Average of Brazilian minimum wage equals BRL 945 (equivalent to USD 100) ^‡^.

**Table 2 pathogens-11-01055-t002:** Prevalence of syphilis and viral co-infections among crack-cocaine users in northern Brazil.

Diagnosis	Positive/Total	% (95% CI)
Syphilis	287/990	29.0 (26.2–32.2)
Co-infections	90/287	31.4 (29.0–34.4)
HBV-TP *	45/287	15.7 (13.0–12.5)
HCV-TP *	17/287	5.9 (2.9–9.5)
HIV-1-TP *	28/287	9.8 (7.2–12.7)
HIV-1-HBV-TP *	11/287	3.8 (0.0–7.1)

* TP: *Treponema pallidum*; HBV: Hepatitis B virus; HCV: Hepatitis C virus; HIV-1: Human immunodeficiency virus type 1.

**Table 3 pathogens-11-01055-t003:** Factors associated with syphilis among people who use crack-cocaine in northern Brazil using logistic regression models.

Factors	Total	Syphilis + (%)	Bivariate OR (95% CI)	Multivariatea OR (95% CI)
18–29 years vs. ≥30 years	632	214 (33.9)	2.0 (1.4–2.8)	2.5 (1.7–4.9)
Up to elementary school vs. High school or more	791	246 (31.1)	1.6 (1.2–2.5)	1.9 (1.4–2.7)
Up to one Brazilian minimum wage vs. More than one minimum wage ^†^	750	242 (32.3)	2.0 (1.4–3.0)	2.4 (1.8–3.2)
Crack-cocaine use ≥40 months vs. Up to 39 months	581	188 (32.4)	1.5 (1.1–2.0)	3.2 (2.6–4.1)
Condomless sex ^†^	875	281 (32.1)	15.2 (6.8–35.6)	18.3 (7.1–31.5)
More than 10 sexual partners ^†^	451	149 (33.0)	1.5 (1.1–1.9)	4.7 (3.5–6.4)
Exchange of sex for money/illicit drug vs. No exchange ^†^	303	115 (38.0)	1.9 (1.4–2.5)	3.6 (2.8–5.1)

^†^ Last 12 months; OR: Odds Ratio; 95% CI: 95% confidence interval; aOR: adjusted Odds Ratio.

## Data Availability

The data presented in this study are available on request from the corresponding author. The data are not publicly available due to the investigation of infections and co-infections with other pathogens.

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
