# Peer review of "Syphilis and Co-Infections with HIV-1, HBV, and HCV among People Who Use Crack-Cocaine in Northern Brazil"

_pathogens, 2022, doi:10.3390/pathogens11091055_

Round 1
Reviewer 1 Report
Authors aimed to evaluate the exposure to Treponema pallidum among crack-cocaine users of northern Brazil.
Authors also intended to evaluate HIV, HBV and HCV infection among people evidencing reactive treponemic serological tests. This solution surely bias evaluation as these infections might be as important among people with non-reactive treponemic tests. It would be of interest to screen for the most common STI but, for some unknown reason, authors ignored Chlamydia trachomatis and Neisseria gonorrhoeae.
Authors state often: “high-risk sexual behavior”, what do authors exactly mean by this?
Authors also state “treat people infected with pathogens” ??? What pathogens? The ones under investigation in the present study?
Pg3: authors state that 30.6% of the participants reported the exchange of sex-for-money or illicit drugs. How many people are these ’30.6%’? This value does not appear in Table 1.
Figure 2 should disappear or go to supplemental
Pg 5: ‘active syphilis’ definition based on serological treponemic tests may bias evaluation as some of these persons might no longer have an ‘active’ infection, or they may have a primary lesion and not yet have developed antibodies.
Discussion is too long. Pg 8, 2nd paragraph “The results record here… pathogen-relevant risk behavior among PWUD [11,13].” should disappear, it is irrelevant. The same for 1st paragraph in pg 9.
Author Response
Dear editor and reviewers,
Most requests and suggestions from reviewers were made, even the title of the manuscript has been changed. All changes made were highlighted in blue in the manuscript. Minor changes were also made to correct errors or facilitate understanding.
Furthermore, all the considerations made by the reviewers were commented by the authors and are located below. The authors are grateful for the attention and contribution of each of the reviewers to a better manuscript.
AUTHORS' RESPONSES TO THE REVIEWER 1:
1) “Authors aimed to evaluate the exposure to Treponema pallidum among crack-cocaine users of northern Brazil. Authors also intended to evaluate HIV, HBV and HCV infection among people evidencing reactive treponemic serological tests. This solution surely bias evaluation as these infections might be as important among people with non-reactive treponemic tests. It would be of interest to screen for the most common STI but, for some unknown reason, authors ignored Chlamydia trachomatis and Neisseria gonorrhoeae”.
Response: The authors agree with reviewer 1 on the importance and need to screen for other pathogens in this vulnerable group. However, currently, we do not have the resources to conduct this search and perform the necessary laboratory tests.
2) “Authors state often: “high-risk sexual behavior”, what do authors exactly mean by this?”.
Response: The authors understand that risky sexual behaviors are behaviors that can affect well-being or even cause serious health problems. They can result from sexual activity with multiple partners, inconsistent use of condom, exchanging unprotected sex for money/drugs, having a partner who uses illicit drugs, and others. The authors changed the term "high-risk sexual behavior" to "risky sexual behavior". New texts in the manuscript: “Long-term crack-cocaine use can greatly increase the risk of acquiring pathogens because PWUCC increasingly engage in risk behaviors and consequently decreased health status and care”; and “In this context, the occurrence of risk sexual behavior is well documented among PWUCC compared to other subpopulations of PWUD [6,11,20,38]”.
3) “Authors also state “treat people infected with pathogens” ??? What pathogens? The ones under investigation in the present study?”.
Response: All study participants received counseling about sexually transmitted infections. Those participants with positive results for syphilis (with or without viral co-infection) were referred to the public health network in those municipalities for follow-up and clinical treatment, according to the protocols of the Brazilian Ministry of Health. The authors did not follow up on the monitoring and clinical treatment of participants with pathogens.
4)“Pg3: authors state that 30.6% of the participants reported the exchange of sex-for-money or illicit drugs. How many people are these ’30.6%’? This value does not appear in Table 1.”.
Response: The requested value was included in the results text and can be seen in table 1. New text in manuscript: “Some participants reported the exchange of sex-for-money or illicit drugs (n = 303; 30.6%)”.
5) “Figure 2 should disappear or go to supplemental”.
Response: The figure 2 was placed in the supplementary material. Now she is Figure S1.
6) Pg 5: ‘active syphilis’ definition based on serological treponemic tests may bias evaluation as some of these persons might no longer have an ‘active’ infection, or they may have a primary lesion and not yet have developed antibodies.
Response: The authors modified the manuscript to avoid doubts in the readers. According to the Brazilian Ministry of Health's technical manual for the diagnosis of syphilis, the detection of treponemal and non-treponemal antibodies is "suggestive of active syphilis". This reference was exposed and used in the new version of the manuscript". New text in manuscript: “According to the technical manual for the diagnosis of syphilis, the detection of treponemal and non-treponemal antibodies is suggestive of active syphilis [46]”.
7) Discussion is too long. Pg 8, 2nd paragraph “The results record here… pathogen-relevant risk behavior among PWUD [11,13].” should disappear, it is irrelevant. The same for 1st paragraph in pg 9.
Response: The authors agree that the discussion has become lengthy, but Pathogens has no restrictions on the size of manuscripts. In addition, the findings of this study are discussed clearly and in depth, so it indicates possible interventions to control and reduce the spread of T. pallidum and take care of PWUCC (including some of these interventions are foreseen in public health policies and were not established in northern Brazil). Historically, the health conditions of the human population residing in northern Brazil (Brazilian Amazon) need to be adjusted, improved, as indicated in the introduction to this manuscript and in other studies [Garnelo, 2019 (https://doi.org/10.1590/0102-311X00220519); Castro, 2022 (https://doi.org/10.1038/s41591-022-01710-9)]. Excluding the paragraphs indicated by the reviewer (facts related to the study findings and which may change the epidemiological scenario) is to agree that such interventions are unnecessary. The authors kindly ask the reviewer 1 and the editor for a greater understanding of these paragraphs, they are connected to the study and point out ways that may change in the future the epidemiological scenario of syphilis and other infections among PWUCC in northern Brazil.
Reviewer 2 Report
Dear Authors,
Thank you for the opportunity to review this paper. This is a wonderful summary of the association between crack cocaine use and syphilis infection among people in the Northeastern Brazilian region surveyed. Please see my suggestions below:
- 1. Please spell out the meaning of the acronym RQT in the abstract when it is first written. It is also important to mention that the RQT is a Trepoemal tests, which could also be described in the methods
- 2. The use of the word “exposure” is inaccurate. A positive RQT indicates that the person has either a prior or current syphilis infection. Exposure itself does not necessitate infection. For example, people can be exposed to syphilis via a sexual partner, but for whatever reason the bacteria is not transmitted from one person to the other). I would change the wording that a positive RQT test shows evidence of lifetime-syphilis, past or present.
- 3.You mention in the discussion that none of those who had a positive RQT were aware of a prior syphilis diagnosis or had a history of treatment. The collection of that data (awareness of syphilis infection) should be included in the methods, and the Results section should include this point as well. It’s very important, as it means 100% of persons with a positive RQT had a syphilis infection that was in need of treatment, and are thus currently infected. This is an astounding finding. It suggests not only a high prevalence of syphilis, but a complete absence of access to diagnosis and treatment. (If I understand this correctly).
- 4. I agree that titers are a proxy for disease burden and sometimes for bacterial load. However, the term “active” syphilis is not entirely accurate. VDRL and RPR titers – particularly those below 1:32 – can vary widely by patient and by HIV status. For instance, late-latent disease and serofast states are also commonly be diagnosed at 1:8. I have seen patients with serofast titers as high as 1:64 (while not common, I have come across this more than once). I’m curious how or why 1:8 was chosen as the cut off. Could you please clarify that? Another approach could be to use what CDC typically considers “high” which is 1:32. Regardless, I would choose a word other than “active”. Perhaps “elevated titers” is more precise.
Thank you for this important work. I look forward to your revisions.
Author Response
Dear editor and reviewers,
Most requests and suggestions from reviewers were made, even the title of the manuscript has been changed. All changes made were highlighted in blue in the manuscript. Minor changes were also made to correct errors or facilitate understanding.
Furthermore, all the considerations made by the reviewers were commented by the authors and are located below. The authors are grateful for the attention and contribution of each of the reviewers to a better manuscript.
AUTHORS' RESPONSES TO THE REVIEWER 2:
“1. Please spell out the meaning of the acronym RQT in the abstract when it is first written. It is also important to mention that the RQT is a Trepoemal tests, which could also be described in the methods”.
Response: The changes requested by reviewer 2 were made to the abstract and materials and methods in the new version of the manuscript. New texts in the manuscript: “Blood samples were collected to assess the presence of Treponema pallidum using the Rapid Qualitative Test (RQT) and the Venereal Disease Research Laboratory (VDRL). Reactive samples by RQT were used to assess the presence of HBV, HCV, and HIV-1 using Enzyme Immunoassay (EIA) and Polymerase Chain Reaction (PCR)”; “RQT for syphilis are considered treponemal tests as they involve the detection of specific antitreponemal antibodies in a sample [46]”; and “VDRL is considered a non-treponemal test, as it detects antibodies that are not specific for T. pallidum, but that are found in patients with syphilis [46]”.
“2. The use of the word “exposure” is inaccurate. A positive RQT indicates that the person has either a prior or current syphilis infection. Exposure itself does not necessitate infection. For example, people can be exposed to syphilis via a sexual partner, but for whatever reason the bacteria is not transmitted from one person to the other). I would change the wording that a positive RQT test shows evidence of lifetime-syphilis, past or present.”
Response: The authors replaced the term “exposure to T. pallidum” by “with syphilis” in the text and highlighted that the term “with syphilis” refers to infection by T. pallidum in the past or present. New text in manuscript: “In this study, PWUCC with non-reactive results by RQT were classified as without syphilis and no procedure or test was performed. All PWUCC with reactive results by RQT were classified as with syphilis (i.e., they have been (past) or are (present) infected with T. pallidum) and had biological samples collected to perform additional tests (Figure 2)”.
“3. You mention in the discussion that none of those who had a positive RQT were aware of a prior syphilis diagnosis or had a history of treatment. The collection of that data (awareness of syphilis infection) should be included in the methods, and the Results section should include this point as well. It’s very important, as it means 100% of persons with a positive RQT had a syphilis infection that was in need of treatment, and are thus currently infected. This is an astounding finding. It suggests not only a high prevalence of syphilis, but a complete absence of access to diagnosis and treatment. (If I understand this correctly)”.
Response: In the new version of the manuscript, knowledge about the presence and possibility of treatment of pathogens such as HBV, HCV, HIV and T. pallidum was included in the materials and methods and results. New texts in the manuscript: “This cross-sectional study relied on biological and self-reported socio-behavioral data (including knowledge about the presence of HBV, HCV, HIV-1, and T. pallidum and treatment of infections caused by these pathogens) from a convenience sample of PWUCC from the municipalities of the state of Amapá and Pará, northern Brazil (Figure 1)”; and “All PWUCCs who tested positive only for syphilis or syphilis and co-infection with HBV, HCV, and/or HIV-1 reported not being aware of their carrier status of these pathogens until participating in this study, consequently, none of them had received treatment for any of the pathogens”.
“4. I agree that titers are a proxy for disease burden and sometimes for bacterial load. However, the term “active” syphilis is not entirely accurate. VDRL and RPR titers – particularly those below 1:32 – can vary widely by patient and by HIV status. For instance, late-latent disease and serofast states are also commonly be diagnosed at 1:8. I have seen patients with serofast titers as high as 1:64 (while not common, I have come across this more than once). I’m curious how or why 1:8 was chosen as the cut off. Could you please clarify that? Another approach could be to use what CDC typically considers “high” which is 1:32. Regardless, I would choose a word other than “active”. Perhaps “elevated titers” is more precise”.
Response: Based on the reviewer's observations, the authors corrected some information and clarified others. The Brazilian Ministry of Health's technical manual for the diagnosis of syphilis was used as a reference (46) for the selection, execution, and interpretation of laboratory tests. The reverse approach (item 4.3 of the manual) was selected. RQT and VDRL were acquired and used. Among the guidelines, we highlight two: i) the sample must be tested pure and diluted to eliminate the possibility of the prozone phenomenon; ii) detection of treponemal and non-treponemal antibodies is suggestive of active syphilis. This information was included in the materials and methods, and the term “active syphilis” was replaced by “suggestive of active syphilis” throughout the manuscript. New texts in the manuscript: “Overall, 287 (29.0%) had reactive results for syphilis. Forty-nine (4.9%) PWUCC had reactive results for treponemal antibodies using Rapid Qualitative Test (RQT) and non-reactive for non-treponemal antibodies using Venereal Disease Research Laboratory (VDRL). Two hundred and thirty-eight (24.0%) PWUCC had reactive results for treponemal antibodies and non-reactive for non-treponemal antibodies, suggestive of active syphilis”; and “VDRL is considered a non-treponemal test, as it detects antibodies that are not specific for T. pallidum, but that are found in patients with syphilis [46]. All samples were tested pure (1:1) and in titrations (³ 1:2) to eliminate the possibility of the prozone phenomenon in the execution of this laboratory test. According to the technical manual for the diagnosis of syphilis, the detection of treponemal and non-treponemal antibodies is suggestive of active syphilis [46]”.
Round 2
Reviewer 1 Report
Authors solved major problems of the manuscript. The discussion remains long and the paragraphs that could disappear (pointed out in the first revision) seem too much politics and not science. Therefore, and despite of the importance of the message to political stakeholders, I would suggest to keep the text within the scientific 'territory'.
Author Response
Historically, epidemiological findings guide public health strategies and policies, as well as record their absence or inefficiency. In the old discussion of this manuscript, three paragraphs contained detailed indications of public health strategies and policies (with indication of scientific references), in force in Brazil for more than a decade (but not used or used only in the metropolitan regions of state capitals in northern Brazil), which should be used throughout the north of Brazil in order to care for and treat PWUCC and reduce the spread of pathogens, such as T. pallidum, HBV, HCV and HIV. In this new version of the manuscript, the discussion was modified as requested by the reviewer. Three paragraphs (from the old discussion) were condensed into just one paragraph, which briefly highlights the interventions needed in the epidemiological scenario.
Paragraph in the new version of the manuscript: "The results also point to interventions in the epidemiological scenario. No PWUCC with syphilis were aware of their infection status and, consequently, were not undergoing any standard treatment [39]. This situation urgently needs to be addressed and improved. It is essential to identify and treat people infected with pathogens, as well as thereby preventing future infections in others. Improvement of resources, services and technical skills must be carried out in public health establishments in this Brazilian region. In addition, the primary risk factor for syphilis is related to sexual risk behaviors, therefore there is a need to improve prevention measures. Targeted education program, provision of resources or prevention materials (e.g., condoms), and a clear understanding of the association of sexual behaviors with other risk behaviors and the economy or acquisition of drugs, and other more specific measures related to the local population are highly important measures of prevention to reduce pathogen-relevant risk behavior among PWUD [11,13]. Finally, interventions should target the issue of crack-cocaine use itself, given the duration and intensity of use of this psychoactive substance in the target population. Most participants require evidence-based treatment for (multi-)drug dependence/disorder [40,41]. The availability of resources to host and care for PWUD are highly limited in northern Brazil [11-13]. The development of intervention measures related to the use of psychoactive substances is very important for the treatment of drug dependence/disorder and to contain or reduce the spread of pathogens in the present target population".
The authors are grateful for the attention and contribution of each of the reviewers to a better manuscript.